# A Theoretical Study of the Occupied and Unoccupied Electronic Structure of High- and Intermediate-Spin Transition Metal Phthalocyaninato (Pc) Complexes: VPc, CrPc, MnPc, and FePc

**DOI:** 10.3390/nano11010054

**Published:** 2020-12-28

**Authors:** Silvia Carlotto, Mauro Sambi, Francesco Sedona, Andrea Vittadini, Maurizio Casarin

**Affiliations:** 1Dipartimento di Scienze Chimiche, Università degli Studi di Padova, via F. Marzolo 1, 35131 Padova, Italy; mauro.sambi@unipd.it (M.S.); francesco.sedona@unipd.it (F.S.); 2Istituto di Chimica della Materia Condensata e di Tecnologie per l’Energia (ICMATE-CNR), via F. Marzolo 1, 35131 Padova, Italy; andrea.vittadini@unipd.it

**Keywords:** transition metal phthalocyaninato complexes, X-ray absorption spectroscopy (XAS), Density Functional Theory (DFT), Restricted Open-Shell Configuration Interaction Singles (ROCIS)

## Abstract

The structural, electronic, and spectroscopic properties of high- and intermediate-spin transition metal phthalocyaninato complexes (MPc; M = V, Cr, Mn and Fe) have been theoretically investigated to look into the origin, symmetry and strength of the M–Pc bonding. DFT calculations coupled to the Ziegler’s extended transition state method and to an advanced charge density and bond order analysis allowed us to assess that the M–Pc bonding is dominated by *σ* interactions, with FePc having the strongest and most covalent M–Pc bond. According to experimental evidence, the lightest MPcs (VPc and CrPc) have a high-spin ground state (GS), while the MnPc and FePc GS spin is intermediate. Insights into the MPc unoccupied electronic structure have been gained by modelling M L_2,3_-edges X-ray absorption spectroscopy data from the literature through the exploitation of the current Density Functional Theory variant of the Restricted Open-Shell Configuration Interaction Singles (DFT/ROCIS) method. Besides the overall agreement between theory and experiment, the DFT/ROCIS results indicate that spectral features lying at the lowest excitation energies (*EE*s) are systematically generated by electronic states having the same GS spin multiplicity and involving M-based single electronic excitations; just as systematically, the L_3_-edge higher *EE* region of all the MPcs herein considered includes electronic states generated by metal-to-ligand-charge-transfer transitions involving the lowest-lying *π*^*^ orbital (7e_g_) of the phthalocyaninato ligand.

## 1. Introduction

Phthalocyanines (H_2_Pc) share with porphyrins (H_2_P), everywhere present “as far as the living world is concerned’’ [1], the same four nitrogen-based coordinative pockets. Even though H_2_Pc and its metal complexes (MPc) are not present in Nature, they have been attracting great interdisciplinary interest because their technological potential spans over a wide range of applications [2,3]. Besides traditional appliances, such as dyestuffs for textiles and inks [3], MPcs are currently used as intrinsic semiconductors, chemical sensors, organic light-emitting diodes, organic photovoltaic cells, thin-film transistors, materials for nonlinear optics, spintronics and laser recording [4,5,6,7,8]. Moreover, bio-inspired oxygen-binding MPcs have been shown as viable substitutes for precious metals in catalysts for the oxygen reduction reaction (ORR) in low-temperature fuel cells [9,10,11,12]. As intimate an understanding as possible of the origin, symmetry, and strength of the M–Pc interaction is then mandatory to enhance the efficiency of new MPc-based devices. As such, X-ray absorption spectroscopy (XAS) is unanimously recognized as a valuable tool to probe, element-selectively, the empty electronic structure of M complexes, the M coordinative environment, as well as the nature and the strength of the M–ligand interaction [13,14,15,16].

Metal L_2,3_-edges’ spectral features are related to the electronic states generated by the electric dipole-allowed 2p → *n*d excitations [17], thus providing information about the contribution of the M-based *n*d atomic orbitals (AOs) to the frontier virtual molecular orbitals (VMOs). Metal-based 2p^6^… *n*d^k^ → 2p^5^… *n*d^k+1^ excitations create a hole in the M 2p core AOs with an angular momentum quantum number *ℓ* = 1, and spin-orbit coupling (SOC) allows $\vec{\ell }$ to couple with $\vec{s}$, whose quantum number *s* = 1/2. Two distinct states are then produced (*j* = 3/2 and *j* = 1/2), with the former (the L_3_-edge) lying at lower excitation energy (*EE*) and having an intensity approximately twice that of the L_2_-edge associated with *j* = 1/2. Besides the M L_2,3_-edges, the ligand (L) donor atom K-edge XA spectra of ML complexes with partly filled *n*d AOs are usually characterized by rather intense pre-edge features. These are associated with the electronic states generated by the electric dipole-allowed L-based 1s → *m*p transitions [17], whose intensity gauges the L *m*p character of frontier VMOs [18,19,20,21]. XAS at the L K-edge thus directly probes the so-called M−L symmetry-restricted covalency [22], affording information complementary to that gatherable by XAS at the M L_2,3_-edges. Despite the fact that the L_2,3_-edges spectra of ML complexes contain a huge amount of chemical information, their first-principle modelling is theoretically demanding because, besides the ligand field and covalency effects, SOC between the possible many final-state multiplets has to be considered [23,24,25,26,27,28,29,30,31,32].

At the very beginning of this century, Koshino et al. [33] recorded the L_2,3_ excitation spectra of MPc (24 ≤ Z ≤ 29; Z corresponds to the atomic number of the metals they considered) by exploiting the inner-shell electron energy-loss spectroscopy (ISEELS); a little over ten years later, Kroll et al. [34] investigated the electronic structure of MPc (25 ≤ Z ≤ 30) by combining soft L-edge XAS and 2p photoemission spectroscopy. In addition, few years ago, Eguchi et al. [35] succeeded in the ultra-high vacuum synthesis and XAS characterization of VPc on Ag(111), while neither XAS nor ISEELS data have been so far reported for TiPc to our knowledge.

As a part of a systematic investigation of the electronic properties of energy-targeted materials, some of us have recently investigated their structure/reactivity relationships by exploiting XAS at the N K-edge and at the M L_2,3_-edges of diverse MPc (M = V [36,37], Fe [12,38] and Cu [37,39,40]) and CuTPP/CuTPP(F) [41] (H_2_TPP = tetraphenylporphyrin; H_2_TPP(F) = tetrakispentafluorophenyl porphyrin) surface-supported films coupled to quantum mechanical calculations. Two different methodologies were adopted to model the M L_2,3_-edges’ features: the current Density Functional Theory variant of the Restricted Open-Shell Configuration Interaction Singles (DFT/ROCIS) method [42] (VPc, FePc and CuPc) and the Relativistic Time-Dependent DFT (RTD-DFT), including SOC with full use of symmetry and correlation effects [43] within the Tamm−Dancoff approximation [44] (CuPc, CuTPP, CuTPP(F)). As such, it has to be noted that, besides L donor atom K-edge XA spectra of ML complexes [39,41,45,46,47], the RTD-DFT approach may be employed to satisfactorily models the M L_2,3_-edges XAS features of closed shell [46,48] and Cu^II^ complexes. Indeed, the modelling of the Cu^II^ L_2,3_-edges features corresponds, among open shell complexes, to the simplest possible case because the electric dipole allowed 2p^6^… 3d^9^ → 2p^5^…3d^10^ transitions generate a final configuration, which has only two term symbols. The corresponding spectral splitting is dominated by the 2p SOC contribution and the overall energetics and intensities are strongly influenced by ligand-field and covalency, respectively [40]. At variance to that, the RTD-DFT approach is unable to suitably describe SOC in open-shell molecules as explicitly reported in the ADF manual. All the MPcs herein considered (M = V, Cr, Mn and Fe) are open-shell systems with a quite complex electronic structure. A RTD-DFT modelling of their XAS features would then be utterly inadequate, while they can be properly handled by exploiting the module ROCIS of the ORCA program package [42]. 

In this contribution, a homogeneous modelling of the VPc, CrPc, MnPc, and FePc L_2,3_-edges’ features is presented and discussed with the ultimate goal of providing a first-principle rationale of differences characterizing the M–Pc interaction along the investigated series. As such, it can be useful to mention that, with the exception of VPc, recently synthesized in extreme conditions [35], the remaining MPcs herein considered all have relevant catalytic applications, including the: (i) CO and NO oxidation as well as ORR (CrPc) [49,50,51,52]; (ii) oxidation reactions in homogeneous and heterogeneous phase (MnPc) [9]; (iii) N-alkylation [53], C–H amination [54], C–C bond formation [55], synthesis of esters [56] and oximes [57] as well as reduction [58], oxidation [59,60], and radical reactions (FePc) [61]. Moreover, bio-inspired oxygen-binding Fe-macrocycles are particularly appealing as a consequence of their ability to reversibly bind O_2_, a crucial step in processes such as respiration, photosynthesis or the ORR catalysis [62]. Thus, there is no doubt that a comprehension as intimate as possible of the M–Pc interaction is of paramount importance to design new MPc-like species devoted to specific purposes.

## 2. Computational Details

The ground state (GS) electronic and structural properties of title molecules have been herein investigated by exploiting the Amsterdam Density Functional (ADF) package [63] within the assumption of an idealized D_4h_ symmetry [17] (see Figure 1), by running spin-unrestricted, nonrelativistic DFT calculations, with generalized gradient corrections self-consistently included through the Becke−Perdew formula [64,65], by adopting a triple-ζ with a polarization function Slater-type basis set for all the atoms, and by freezing the M 1s–2p AOs and the 1s AO of N and C atoms throughout the calculations. MPc optimized Cartesian coordinates are reported in Appendix A of the Appendix A.

Insights into the origin, symmetry, and strength of the M–Pc interaction have been gained by combining the Nalewajski−Mrozek approach [66,67,68,69,70,71], well suited to estimating bond multiplicity indices (^NM^I) in M complexes [66,67,68,69,70,71,72], with the Ziegler’s extended transition state (ETS) method [73]. According to the ETS scheme, the MPc bonding energy (*BE*) may be written as:(1)$$BE=-\left(\Delta E_{es}+\Delta E_{Pauli}+\Delta E_{orb}\right)$$
where Δ*E_es_* accounts for the pure electrostatic interaction, Δ*E_Pauli_* represents the destabilizing two-orbital−four-electrons interaction between the occupied orbitals of the interacting fragments (only atomic fragments have been herein considered), and Δ*E_orb_* corresponds to the stabilizing interaction between the occupied and empty orbitals of the atomic fragments. In passing, Δ*E_orb_* may be further decomposed into contributions due to the different irreducible representations (IRs) of the D_4h_ point group, according to
(2)$$\Delta E_{orb}=\sum_{\chi }\Delta E_{orb}^{\chi },$$

The MPc L_2,3_-edges’ XA spectra [33,34,35] have been modelled by evaluating *EE*s and corresponding oscillator strength distributions (*f*(*EE*)) for transitions with the M 2p-based MOs as initial spin orbitals (*isos*), by means of the DFT/ROCIS method [30], which includes SOC in a molecular Russell−Saunders fashion [25], by adopting the B3LYP exchange–correlation (XC) functional [74] for VPc, CrPc and FePc and the M06 meta-GGA XC [75] for MnPc (vide infra), and by using the def2-TZVP(-f) basis set [76,77]. The combined use of DFT and configuration interaction requires a set of three semi-empirical parameters (c_1_ = 0.18, c_2_ = 0.20, and c_3_ = 0.40), which have been calibrated by Roemelt and Neese [25] for a test set of M L_2,3_-edges. Throughout the M L_2,3_-edges modelling, the resolution of identity approximation has been used with the def-TZVP/J basis set [76,77]. Moreover, the zero-th order regular approximation has been adopted to treat the scalar relativistic effects [78]. Numerical integrations for DFT/ROCIS calculations have been carried out on a dense Lebedev grid (302 points) [79]. In addition, MPc-modelled spectra have been shifted by 10.9 (VPc), 9.8 (CrPc), 8.3 (MnPc) and 13.1 (FePc) eV to superimpose the highest intensity features of the simulated and experimental L_3_-edge, which does not suffer from the extra broadening and the distortion due to the Coster–Kronig Auger decay process [32,80]. This was needed because absolute theoretical *EE*s carry errors arising from DF deficiencies in the core region, one-particle-basis set restrictions and inadequacies in the modelling of spin-free relativistic effects [24].

## 3. Results and Discussion

### 3.1. MPc Occupied Electronic Structure

The aim of obtaining an understanding as intimate as possible of the origin, symmetry and strength of the M–Pc interaction may benefit from a preliminary, qualitative description of the MPc frontier orbitals simply based on symmetry arguments and overlap considerations. MPcs are united by the presence of the Pc^2−^ ligand whose electronic properties have been thoroughly described elsewhere [81]. Pc^2−^ frontier MOs may be split into *σ* and *π* sets. MPc symmetry adapted linear combinations (SALC) of C and N 2p*_σ_* (C and N 2p*_π_*) are bases for the following D_4h_ IRs: a_1g_, a_2g_, b_1g_, b_2g_, e_u_ (e_g_, a_1u_, a_2u_, b_1u_, b_2u_); moreover, among *π* MOs, no a_1u_ SALC of N 2p*_π_* AOs, no b_1u_ SALC of N^Py^ 2p*_π_* AOs and no b_2u_ SALC of N^m^ 2p*_π_* AOs (see Figure 1) is present. In addition, the four N^Py^ lone pairs pointing toward the centre of the coordinative pocket are bases for the a_1g_, b_1g_ and e_u_ IRs. In more detail, the two Pc^2−^ highest occupied MOs (HOMOs) correspond to the 15b_1g_ MO, *σ* in character and strongly localized on the N^Py^ lone pairs pointing toward the centre of the coordinative pocket, and the 2a_1u_
*π* MO. In this regard, it is noteworthy that the D_4h_ a_1u_ IR is anti-symmetric with respect to the reflections through the σ_h_, σ_v_ and σ_d_ symmetry planes of the D_4h_ point group; the a_1u_
*π* MOs have then a node on symmetry planes passing through N^Py^ and N^m^ atoms (see Figure 1). As far as the D_4h_ Pc^2−^ lowest unoccupied MO (LUMO) is concerned, the 6e_g_ VMO has a *π* character too. 

The presence of the Pc^2−^ square planar ligand field lifts the five-fold degeneracy of the M 3d AOs, generating a 3d*_σ_* and a 3d*_π_* set. The former set includes the 21a_1g_ (z^2^-based) and the 16b_1g_ (x^2^–y^2^-based) MOs, while the latter takes in the 14b_2g_ (xy-based) and the 6e_g_ (xz- and yz-based) ones [17]. Among them, the 21a_1g_ and the 14b_2g_ MOs are substantially M–N^Py^ non-bonding, while the 6e_g_ and 16b_1g_ MOs are M–N^Py^
*π* and *σ* antibonding, respectively. Relative energy positions of the 3d*_σ_*/3d*_π_* spin up (↑)/spin down (↓) sets in VPc, CrPc, MnPc and FePc are displayed in Figure 2, together with those of selected Pc-based MOs.

*VPc ground state*. Experimental [35] and theoretical [36,37,82] results disagree about the VPc ground state (GS) spin multiplicity. Carlotto et al. [36,37] recently proposed a ^4^E_g_ high-spin (HS) GS ($a_{1g}^{1}\;b_{2g}^{1}\;e_{g}^{1}\;b_{1g}^{0}$; see Figure 2 and Table 1; possible Jahn–Teller distortions [83] associated with orbitally degenerate GS or excited states have not been taken into account), while Eguchi et al. [35] presumed a ^2^E_g_ low-spin (LS) GS ($a_{1g}^{0}\;b_{2g\;}^{2}e_{g}^{1\;}\;b_{1g}^{0}$). The ^2^Eg state generated by the $a_{1g}^{0}b_{2g}^{2}e_{g}^{1}b_{1g}^{0}$ configuration is 52.1 (35.5) kcal/mol less stable than the HS ^4^E_g_ (LS ^2^B_1g_) one; moreover, the ^4^B_1g_/^4^A_2g_ states generated by the constrained $a_{1g}^{0}b_{2g}^{1}e_{g}^{2}b_{1g}^{0}/a_{1g}^{1}b_{2g}^{0}e_{g}^{2}b_{1g}^{0}$ configurations are 1.8/1.7 kcal/mol less stable than the ^4^Eg GS. Consistently with the VPc $a_{1g}^{1}\;b_{2g}^{1}\;e_{g}^{1}\;b_{1g}^{0}$ GS configuration, the ^NM^I_V-N_^Py^ is quite large (0.64) [36]; as such, since ^NM^I includes both covalent and ionic contributions, it is of some relevance to mention that the V Hirshfeld [84] charge (Q_V_) amounts to 0.34. 

A closer look at the frontier VPc GS electronic structure indicates that all the V 3d-based singly occupied MOs (SOMOs) lie well above the ring-based, V-free, 2a_1u_
*π* doubly occupied MO (DOMO). The ionization energies (*IE*s) of VPc frontier MOs are not available in the literature; nevertheless, their values may be estimated by exploiting the Slater transition state (TS) method [85], which allows the evaluation of excitation/ionization energies “… by means of an artificial state that is halfway between the ground state of an atom or molecule and an excited state” [86]. Interestingly, the lowest VPc TS*IE* (6.17 eV) is associated with the ionization from the V 3d*_π_*-based 6eg^↑^ SOMO rather than with the photoemission from the ring-based, V-free, 2a_1u_
*π* DOMO (6.59 eV). In this regard, it is of value to highlight that in his seminal paper devoted to the investigation of gas-phase photoelectron (PE) spectra of H_2_Pc and MPc (M = Mg, Fe, Co, Ni, Cu, and Zn), J. Berkowitz pointed out that ‘‘… the first ionization potential occurs at ~6.4 eV, and it varies almost imperceptibly from sample to sample, including metal free and MgPc. Therefore, the conclusion seems inescapable that the first ionization potential corresponds to electron ejection from a ring orbital, and not a metal orbital’’ [87].

The comparison of the VPc bond lengths and bond angles with those optimized for the other MPc (see Table 2) [88] indicates that the structural perturbations induced by the presence of different M^II^ ions in the Pc^2-^ coordinative pocket are rather minute.

*CrPc ground state*. As already mentioned, CrPc has been attracting great interest as a catalyst for the CO and NO oxidation, as well as for the ORR [49,50,51,52], thus making particularly interesting the study of its electronic structure. Cr^II^ has a 3d^4^ configuration, which may generate three spin states with S = 0 (LS), S = 1 (intermediate spin; IS) and S = 2 (HS). Any attempt to optimize the LS state failed (NC, non-converged in Table 1), while the ^3^E_u_ IS state, associated with the $a_{1g}^{1}b_{2g}^{1}e_{g}^{2}b_{1g}^{0}2a_{1u}^{0}$^↑^/$a_{1g}^{0}b_{2g}^{0}e_{g}^{1}b_{1g}^{0}2a_{1u}^{1}$^↓^ configuration, has been found less stable than the ^5^B_1g_ HS one, generated by the $a_{1g}^{1}b_{2g}^{1}e_{g}^{2}b_{1g}^{0}$ configuration, by 25.6 kcal/mol (IS and HS CrPc optimized structures are perfectly superimposable). Incidentally, the ^3^E_u_ IS state implied a non-Aufbau occupation accompanied by a pseudo reduction (oxidation) of the Cr^II^ ion (macrocycle). As such, even though the ^5^B_1g_ HS GS has been experimentally revealed [89,90] and theoretically predicted [36,82], the localization of VMOs is still controversial. Indeed, SIESTA [91] numerical experiments carried out by Arillo-Flores et al. [82] are consistent with the absence of “… metal contributions to HOMO and LUMO, they principally localize upon the inner ring.”, which is certainly correct for the M-free 2a_1u_ HOMO, but wrong for the 6e_g_^↓^ LUMO. In addition, differently from VPc, the ring-based, Cr-free, the 2a_1u_
*π* DOMO corresponds to the CrPc HOMO (see Figure 2). Analogously to VPc, the *IE*s of the CrPc frontier MOs are not available in the literature, but, differently from VPc, the lowest TS*IE* value (6.60 eV) is estimated for the ionization from the ring-based, Cr-free, 2a_1u_
*π* HOMO.

Upon moving from VPc to CrPc, the GS frontier electronic configuration evolves from $a_{1g}^{1}b_{2g}^{1}e_{g}^{1}b_{1g}^{0}$ to $a_{1g}^{1}b_{2g}^{1}e_{g}^{2}b_{1g}^{0}$. The addition of an electron to the 3d*_π_*-based 6e_g_^↑^ MO, M–N^Py^ anti-bonding could then be invoked to rationalize the ^NM^I_M-N_^Py^ reduction from 0.64 (see above) to 0.43. Nevertheless, three things need to be kept in mind before drawing conclusions: (i) as already mentioned, ^NM^I includes both covalent and ionic contributions; (ii) Q_Cr_ (0.49) is larger than Q_V_ (0.34); (iii) the 3d*_π_*-based 6e_g_^↑^ MO is more anti-bonding in CrPc than in VPc (the localization % of the VPc 6e_g_^↑^ MO on N^Py^ is negligible; see Figure 3).

Analogously to VPc, no crystallographic data are available in the literature for CrPc; nevertheless, the tiny differences between the CrPc- and VPc-optimized structural parameters seem to indicate that a subtle balance between ionic and anti-bonding covalent contributions to the M–Pc interaction, both of them larger in CrPc than in VPc, takes place.

*MnPc ground state*. Likewise CrPc, LS, IS and HS states are possible; moreover, even though the optimized structural parameters corresponding to different spin states are very similar, the ^4^E_g_ IS state associated to the $a_{1g}^{1}b_{2g}^{1}e_{g}^{3}b_{1g}^{0}$ configuration (see Figure 2) has been found more stable than the ^6^E_g_ HS and the ^2^B_2u_ LS ones by 16.5 and 14.9 kcal/mol, respectively. As such, it has to be noted that: (i) the MnPc IS GS has been experimentally [94,95,96,97,98,99] and theoretically [36,100,101,102,103,104,105] assessed; (ii) the ^6^E_g_ HS state is generated by the $a_{1g}^{1}b_{2g}^{1}e_{g}^{2}b_{1g}^{0}7e_{g}^{1}$ configuration; (iii) the ^2^B_2u_ LS state has the following occupation numbers: $a_{1g}^{1}b_{2g}^{1}e_{g}^{2}b_{1g}^{0}2a_{1u}^{0}$^↑^/$a_{1g}^{0}b_{2g}^{0}e_{g}^{2}b_{1g}^{0}2a_{1u}^{1}$^↓^. In this contest, it is noteworthy that: (i) both the 7e_g_ and the 2a_1u_ MOs are ring-based *π* orbitals, so that the HS (LS) state would imply a pseudo Mn^II^ oxidation (reduction) with the Mn 3d-based orbitals occupied by four (six) electrons (^HS^Q_Mn_, ^IS^Q_Mn_ and ^LS^Q_Mn_ amount to 0.41, 0.33 and 0.28, respectively); (ii) to our knowledge, only two contributions [33,106] suggested a MnPc HS state. As regards the MnPc IS GS, a further controversy concerns its symmetry, or, equivalently, the occupation numbers of the Mn 3d-based 21a_1g_, 14b_2g_ and 6e_g_ MOs. Three different configurations may be considered: $a_{1g}^{1}b_{2g}^{1}e_{g}^{3}b_{1g}^{0}$ (^4^E_g_), $a_{1g}^{1}b_{2g}^{2}e_{g}^{2}b_{1g}^{0}$ (^4^A_2g_) and $a_{1g}^{2}b_{2g}^{1}e_{g}^{2}b_{1g}^{0}$ (^4^B_1g_). In agreement with the theoretical results of Brumboiu et al. [105], the ADF outcomes rule out the ^4^B_1g_ state because of its high energy; moreover, it is noteworthy that different experimental studies support either a ^4^E_g_ or a ^4^A_2g_ GS. Specifically, XAS evidence [98], magnetic circular dichroism (MCD)/UV-Vis results [95], and XAS/MCD outcomes [99] favour a ^4^E_g_ GS [107], while magnetic susceptibility measurements have been rationalized within the assumption of a ^4^A_2g_ GS [95,97], which has been attributed to intermolecular interactions in the MnPc crystal. In agreement with the literature [101], the ADF results herein reported estimate the ^4^E_g_ state to be more stable than the ^4^A_2g_ by 7.3 kcal/mol.

Before going on, it is of some relevance to point out that the MnPc $a_{1g}^{1}b_{2g}^{1}e_{g}^{3}b_{1g}^{0}$ GS configuration implies the presence of a high-lying Mn 3d*_π_*-based SOMO^↓^ well above the ring-based, Mn-free, 2a_1u_
*π* DOMO (see Figure 2). As such, no gas-phase photoemission results are available in the literature for MnPc; however, Grobosch et al. [108] were able to record the He(I) photoemission spectrum of an MnPc thin film deposited on polycrystalline Au. Interestingly, they assigned the lowest lying peak of the MnPc valence band photoemission spectrum to the ionization from the Mn 3d*_π_*-based 6e_g_^↓^ MO, observing also that the Δ*IE* between the ring-based, Mn-free, *π* 2a_1u_ DOMO and the 3d*_π_*-based 6e_g_^↓^ orbital is ~0.5 eV [108,109]. In perfect agreement with these findings [108], the TS*IE*s [85] of the highest-lying 6e_g_^↓^ and 2a_1u_^↓^ orbitals are 5.99 and 6.51 eV, respectively.

Among the investigated molecules, MnPc is the lightest one for which structural parameters are available [88,92]. The data reported in Table 2 reveal that optimized bond lengths and bond angles fairly reproduce experimental evidence. In this context, the ^NM^I_Mn-N_^Py^ value (0.52), just in between the ^NM^I_Cr-N_^Py^ and the ^NM^I_V-N_^Py^ ones (0.43 and 0.64, respectively), seems to indicate that the M–N^Py^ bond-weakening associated with the addition of a further electron to the 3d*_π_*-based 6e_g_ MO is negligible (see Appendix A of the Appendix A). In addition, it has to be underlined that Q_Mn_ (0.33) and Q_V_ (0.34) are almost identical.

*FePc ground state.* Similarly to MnPc, experimental [34,110,111,112,113,114] and theoretical [38,104,115] evidence indicates a FePc IS GS whose symmetry is, however, still debated. On the computational side, the IS GS symmetry, inextricably linked to the occupation numbers of frontier MOs, has been found to be extremely sensitive to the adopted XC functional and basis set. Carlotto et al. [38] have proposed a ^3^E_g_ IS GS, generated by a $a_{1g}^{1}b_{1g}^{2}e_{g}^{3}b_{2g}^{0}$ configuration, on the basis of numerical experiments carried out by employing the ORCA program package [42], by using the hybrid B3LYP XC–functional [74], by adopting the def2-TZVP(-f) basis set [76,77] and the c_1_, c_2_, and c_3_ semi-empirical parameters 0.21, 0.49, and 0.29 (hereafter, old set), respectively. In passing, the $a_{1g}^{1}b_{1g}^{2}e_{g}^{3}b_{2g}^{0}$ GS configuration is a consequence of the orientation of the molecule in the xy plane. The x and y axes of the framework they adopted point toward the N^m^ atoms rather than toward the N^Py^ ones (see Figure 1). The four N^Py^ lone pairs were then bases for the a_1g_, b_2g_ and e_u_ IRs rather than for the a_1g_, b_1g_ and e_u_ ones, and the Fe 3d-based VMO accounting for the Fe-N^Py^
*σ* anti-bonding interaction corresponded to the 14b_2g_ level rather than to the 16b_1g_ one. All the possible configurations compatible with a FePc IS GS have been herein considered. As such, even though the ADF ^3^B_2g_ ($a_{1g}^{1}b_{2g}^{1}e_{g}^{4}b_{1g}^{0}$) and the ^3^E_g_ ($a_{1g}^{1}b_{2g}^{2}e_{g}^{3}b_{1g}^{0}$) IS states are less stable than the ^3^A_2g_ one ($a_{1g}^{2}b_{2g}^{2}e_{g}^{2}b_{1g}^{0}$) by minute amounts (1.4 and 1.1 kcal/mol, respectively), it is of some relevance to underline that the $\Delta E_{orb}^{\chi }$ (χ = a_1g_, b_2g_ and e_g_) of the IS states differ by up to ~230 kcal/mol ($\Delta E_{orb}^{e_{g}}$ = −2584.4, −2810.4 and −2690.4 kcal/mol in the ^3^A_2g_, ^3^B_2g_ and ^3^E_g_ states, respectively). A further ADF ^3^E_g_ state may be generated by the $a_{1g}^{2}b_{2g}^{1}e_{g}^{3}b_{1g}^{0}$ configuration. Besides the non-Aufbau filling of the corresponding electronic levels, the latter ^3^E_g_ state is significantly less stable than the ^3^A_2g_ GS (12.3 kcal/mol). Additional numerical experiments have been carried out to estimate the FePc *BE* of the LS and HS states. As far as the former is concerned, it may imply either the $a_{1g}^{0}b_{2g}^{2}e_{g}^{4}b_{1g}^{0}$ configuration or the $a_{1g}^{2}b_{2g}^{0}e_{g}^{4}b_{1g}^{0}$ one. The LS $a_{1g}^{2}b_{2g}^{0}e_{g}^{4}b_{1g}^{0}$ frozen configuration generates a non-Aufbau energy level filling, and a *BE* lower (~32 kcal/mol) than that of the ^3^A_2g_ GS; any attempt to get a converged *BE* value for the LS $a_{1g}^{2}b_{2g}^{0}e_{g}^{4}b_{1g}^{0}$ frozen configuration failed. Analogous considerations hold for the HS state.

FePc is the lightest MPc for which gas-phase photoemission spectroscopy data have been recorded [87]. According to experimental evidence, the TS*IE* of the ring-based, Fe-free, *π* 2a_1u_ DOMO is the lowest one. Incidentally, its value (6.51 eV) is very close to the one (6.49 eV) estimated by Liao and Scheiner [101], even though a non-relativistic approach has been herein adopted.

According to the experiment in Ref. [93], the shortening of the M–N^Py^ distance upon moving from MnPc to FePc is correctly reproduced (see Table 2). In this context, it is of relevance to mention that Q_Fe_ = 0.13; i.e., the smallest value along the investigated series. Such a result, coupled to the ^NM^I_Fe-N_^Py^ = 0.64, ultimately indicates that the Fe–N^Py^ interaction is the most covalent among the molecules we took into account (look at the MPc Δ*E_orb_* values reported in Table 3). In addition, the presence of a strong Fe–N^Py^ covalent *σ* interaction is consistent with the high-energy position of the 16b_1g_^↑^ VMO (see Figure 2), which accounts for the *σ* anti-bonding interaction between the Fe 3d_x_^2^_-y_^2^ AO and the b_1g_ SALC of the N^Py^ lone pairs.

### 3.2. MPc Unoccupied Electronic Structure

The MPc L_2,3_-edges’ XA spectra [33,34,35,38] herein modelled have been collected from deposits of different thicknesses, often consisting of randomly oriented and weakly interacting molecules. Thus, according to a procedure successfully tested in the past [36,37,38,39,40,41,116,117,118,119,120], XAS outcomes have been rationalized by completely neglecting the adsorbate/substrate interactions. Now, before moving to the modelling of the MPc XAS features, a few words about the 2p → 3d excitations simply based on symmetry arguments may be useful to facilitate the forthcoming discussion. In a simplified picture of the 2p → 3d one-electron excitations, the M^II^ electronic configuration moves from the starting …2p^6^…3d^k^ to the ending …2p^5^…3d^k+1^ with 3 ≤ k ≤ 6 for 23 ≤ Z ≤ 26. The electronic states associated to the …2p^5^…3d^k+1^ configurations, straightforwardly obtained by evaluating the ^2^P ⊗ D^k+1^ direct products (the whole set of multiplets arising from a particular 3d^k+1^ configuration is herein collectively labelled D^k+1^) [121], are collected in Appendix A of the Appendix A. The ligand-field, covalent interactions and SOC admixture further split them, to generate totals of 6 × 210 = 1260 (k + 1 = 4), 6 × 252 = 1512 (k + 1 = 5), 6 × 210 = 1260 (k + 1 = 6), and 6 × 120 = 720 (k + 1 = 7) molecular magnetic spin sublevels, respectively. The M^II^ electronic configurations of the MPc herein considered (HS V^II^
$a_{1g}^{1}b_{2g}^{1}e_{g}^{1}b_{1g}^{0}$, HS Cr^II^
$a_{1g}^{1}b_{2g}^{1}e_{g}^{2}b_{1g}^{0}$, IS Mn^II^
$a_{1g}^{1}b_{2g}^{1}e_{g}^{3}b_{1g}^{0}$, IS Fe^II^
$a_{1g}^{2}b_{2g}^{2}e_{g}^{2}b_{1g}^{0}$) allow us to foresee that the one-electron excitation patterns describing the M^II^ final states in the D_4h_ symmetry should in principle include states having a spin multiplicity either equal to (ΔS = 0) or lower/higher than (ΔS = ±1) the GS. The ΔS = 0 spin-selection rule is slightly released when SOC is considered; more specifically, SOC connects the terms with resultant spins S and S′, where |S − S′| = 0, 1 [122].

Electric dipole-allowed transitions imply that [17]
Γ_GS_ ⊗ Γ*_μ_* ⊗ Γ_ES_ ⊃ Γ_Sym_,(3)
where Γ_GS_, Γ*_μ_*, Γ_ES_ and Γ_Sym_ correspond to the IRs of the MPc electronic GS, the dipole moment operator (a_2u_ + e_u_) [17], the electronic excited state (Γ_ES_ = Γ*_iso_* ⊗ Γ_GS_ ⊗ Γ*_fso_*; *iso* and *fso* stand for initial and final spin orbitals, respectively), and the totally symmetric representation of the D_4h_ point group (a_1g_), respectively. Equation (3) may then evolve to
Γ*_iso_* ⊗ (a_2u_ + e_u_) ⊗ Γ*_fso_* ⊃ a_1g_,(4)
which implies that, within the adopted approximation, which reduces the complete one-electron excited configuration space (1h–1p space) to the subspace where only the M 2p core electrons (transforming as a_2u_ + e_u_) [17] are excited, the allowed electric dipole transitions are
(a_2u_ → a_1g_)^⊥^(5)
(a_2u_ → e_g_)^||^(6)
(e_u_ → e_g_)^⊥^(7)
(e_u_ → a_1g_/a_2g_/b_1g_/b_2g_)^||^(8)
where the ||/⊥ symbols stand for parallel/perpendicular to the molecular *σ*_h_ plane (see Figure 1).

*VPc L_3_-edge.* To date, only Eguchi et al. have succeeded in synthesizing, even though in extreme conditions, surface-supported mono- and multi-layers of VPc, whose angle-dependent linearly polarized XA spectra at the V L_2,3_-edges are reported in Ref. [35]. As such, a thorough analysis of the || and ⊥ V L_2,3_-edges components of the VPc and OVPc XA spectra has been recently reported by Carlotto et al. [36,37]. With specific reference to the VPc complex, the authors were able to conclusively assess its spin state by modelling corresponding XAS features for both LS (S = 1/2) and HS (S = 3/2) states. A brief description of their HS results [36] is herein included to favour the comparison with the modelled spectra of diverse MPcs.

It has been already mentioned that the one-electron excitation pattern describing the V^II^ final states in D_4h_ symmetry should be dominated by states which may have (see Appendix A of the Supporting Materials) either a spin multiplicity equal to (ΔS = 0) or lower/higher (ΔS = ±1) than the GS one [122]. As such, it has to be underlined that the ORCA B3LYP HS GS corresponds to the ^4^A_2g_ state generated by the $a_{1g}^{1}b_{2g}^{0}e_{g}^{2}b_{1g}^{0}$ electronic configuration. B3LYP/ROCIS outcomes [36,37] indicate that both the ^||^*f*(*EE*) and the ^⊥^*f*(*EE*) distributions of the L_3_-edge mainly arise from states having ΔS = 0, while ΔS = ± 1 contributions are negligible. Moreover, the lowest-lying ^||^^/^^⊥^L_3_^1^ features (see Figure 4a) are due to states generated by V 2p → SOMOs single electronic excitations. Incidentally, SOMOs correspond to the V 3d-based 6e_g_ and 21a_1g_ orbitals, while states associated to coupled-single electronic excitations [25,32], mainly involving V 2p → SOMOs and SOMOs → VMOs excitations, contribute to ^||^^/^^⊥^L_3_^2^. SOMOs naturally correspond to the 21a_1g_ and 6e_g_ MOs, while the VMOs are the V 3d-based 14b_2g_ (~90%) and the *π*^*^ Pc-based 7e_g_ (~10%) orbitals. In passing, the MPc 7e_g_ VMO corresponds to the lowest-lying Pc-based *π*^*^ orbital and metal-to-ligand-charge-transfer (MLCT) transitions in diverse MPc L_3_-edge spectra (vide infra) involve this VMO. No contribution from V 2p → 16b_1g_ excitations is provided to states associated with the VPc L_3_-edge. 

The agreement between the experimental evidence and B3LYP/ROCIS outcomes has been documented in detail elsewhere [36,37]; here, it is sufficient to underline that both the relative positions and relative intensities of spectral features are well reproduced by ^||^^/^^⊥^*f*(*EE*). Major disagreements between theory and experiment usually affect the L_2_ region [31]; nonetheless, it deserves mentioning that the B3LYP/ROCIS ^||^L_2_^1^–^||^L_3_^2 HS^Δ*EE* (6.4 eV, see Figure 4a) fairly reproduces the experimental value (6.9 eV). Any further comment about the VPc L_2_-edge is herein avoided.

*CrPc L_3_-edge.* To date, the only CrPc *f*(*EE*) distribution for the 2p excitations is the one recorded by Koshino et al. by exploiting ISEELS [33]. Their spectrum includes both the L_3_- and the L_2_-edge, but the coarse *EE* scale they adopted prevents the possibility of revealing the presence of possible structures associated to them; moreover, no information is provided by the authors about the L_2_–L_3_ Δ*EE*. Similarly to VPc, the one-electron excitation pattern describing the Cr final states in the CrPc D_4h_ symmetry is dominated by states which may have (see Appendix A of the Supporting Material) a spin multiplicity either equal to (ΔS = 0) or lower/higher (ΔS = ±1) than the GS one. The HS CrPc ^||^*f*(*EE*) and ^⊥^*f*(*EE*) B3LYP/ROCIS distributions are superimposed upon the CrPc ISEEL spectrum in Figure 4b. In the L_3_-edge region, ^||^^/^^⊥^*f*(*EE*) consists of an intense peak (L_3_^1^) with an evident shoulder on its higher *EE*s (L_3_^2^, Δ*EE* = 1.40 eV). Moreover, the B3LYP/ROCIS outcomes also indicate that both ΔS = 0 and ΔS = −1 states, both of them associated with single electronic excitations, contribute to the ^||^^/^^⊥^*f*(*EE*) distributions. In more detail, ^||^^/^^⊥^L_3_^1^ features are caused by ΔS = 0 states generated by Cr 2p → SOMOs transitions, with the whole set of the half-occupied Cr 3d-based orbitals (6e_g_, 21a_1g_ and 14b_2g_ SOMOs) as *fsos*. At variance with that, the ^||^^/^^⊥^L_3_^2^ shoulders include contributions from both ΔS = 0 (64%) and ΔS = −1 (30%) states. Quintet states have the same origin as those associated with ^||^^/^^⊥^L_3_^1^, while the triplet ones are related to Cr-based 2p → 16b_1g_ and MLCT 2p → *π*^*^ Pc-based transitions. As such, it is noteworthy that states associated with the Cr-based 2p → 16b_1g_ transition only contribute to ^||^L_3_^2^. Analogously to VPc, any comment about the CrPc L_2_-edge is herein avoided as it cannot be unambiguously determined by experiment [32]; nevertheless, we underline that the B3LYP/ROCIS L_2_–L_3_^1 HS^Δ*EE* (8.6 eV, see Figure 4b) fairly reproduces the experimental value (7.9 eV).

*MnPc L_3_-edge.* The MnPc L_2,3_-edges’ XA spectrum recorded by Koshino et al. [33] suffers from the same issues already mentioned for CrPc. Otherwise, the experimental evidence reported by Kroll et al. [34] for MPc (25 ≤ Z ≤ 30) is much more informative as a consequence of the *EE* range they showed for each single MPc, thus allowing the detection of the fine structure eventually contributing to spectral features; moreover, both || and ⊥ polarized XA spectra are reported for each MPc. In the forthcoming discussion, ISEELS outcomes of Koshino et al. will no longer be considered as a reference for the L_2,3_-edges XAS modelling herein presented. At least three well-evident and closely spaced peaks (herein labelled ^||^L_3_^1^, ^||^L_3_^2^ and ^||^L_3_^3^, and lying at ~640.0, ~641.0 and ~643.0 eV, respectively) contribute to the MnPc ^||^L_3_-edge spectrum (see Figure 4c). In addition, the ^||^L_3_^3^’s higher *EE* side is characterized by the presence of an evident shoulder at ~644.0 eV. The || → ⊥ light polarization switching is accompanied by a significant relative intensity reduction in spectral features having *EE*s in between 642 and 644 eV, thus reducing the ^⊥^L_3_-edge spectrum to the ^⊥^L_3_^1^ and ^⊥^L_3_^2^ peaks with comparable intensity and lying at ~640.5 and ~641.5 eV, respectively (see Figure 4c).

The MnPc B3LYP/ROCIS ^||^^/^^⊥^*f*(*EE*) distributions, not herein included, poorly reproduces the L_3_-edge XA spectrum in terms of number of peaks, relative energy positions and relative intensities. A few years ago, Carlotto et al. [123] tested the efficiencies of several diverse XC functionals (non-hybrid, hybrid and hybrid meta-GGA) in reproducing the L_2,3_-edges’ absorption spectra of Mn complexes. The use of the hybrid M06 meta-GGA XC functional [75] was found to be decisive for a detailed assignment of the Mn(acac)_3_ (acac = acetylacetonato) L_2,3_-edges XAS features. Now, despite the inability even of the M06 XC functional to reproduce in detail the complex structure of the MnPc XA spectrum, in particular the presence of the closely spaced ^||^L_3_^1^ and ^||^L_3_^2^ peaks, the IS-corresponding modelling (see Figure 4c) provides a satisfactory agreement between experiment and theory. In more detail, theoretical results indicate that the lowest-lying feature of the ^||^*f*(*EE*) distribution (^||^L_3_^1^, representative of the ^||^L_3_^1^ and ^||^L_3_^2^ experimental peaks; see Figure 4c) has to be associated to the quartet electronic states (ΔS = 0) generated by single electronic excitations having the whole Mn 2p set as *isos* and the low-lying Mn 3d-based 21a_1g_^↓^, 14b_2g_^↓^ and 6e_g_^↓^ VMOs as *fsos*. At variance with that, electronic states with ΔS = 0 (58%) and ΔS = 1 (24%) contribute to the intense ^||^L_3_^3^ feature, while electronic states with ΔS = -1 negligibly contribute to the ^||^^/^^⊥^L_3_ patterns. Former states (ΔS = 0) imply Mn 2p → SOMOs (21a_1g_, 14b_2g_ and 6e_g_) and SOMOs → 16b_1g_/7e_g_ coupled-single excitations [25,32], while the latter (ΔS = 1) are mainly generated by Mn 2p → *π*^*^ Pc-based single electronic excitations. The ΔS = 1 Mn 2p → *π*^*^ Pc-based 4b_2u_ single electronic excitation violates selection rules stated by Eqns 5–8. As such, it has to be mentioned that i) ORCA calculations have to be run in C_1_ symmetry and ii) the *f* value of the ΔS = 1 Mn 2p → *π*^*^ Pc-based 4b_2u_ single electronic excitation is very low.

The satisfactory agreement between ^||^^/^^⊥^*f*(*EE*) distributions and experimental evidence [34] prompts us to further detail the proposed assignment. Despite the fact that electronic states generating the ^||^^/^^⊥^L_3_^1^ features of Figure 4c are associated with Mn-based ΔS = 0 2p → 3d single electronic excitations involving the whole 2p set and the low-lying Mn 3d-based ↓ VMOs, it sounds reasonable that the (1a_2u_ → 21a_1g_)^⊥^ and (1e_u_ → 6e_g_)^⊥^ transitions contribute to the L_3_ lower *EE* region more than the (1a_2u_ → 6e_g_)^||^, (1e_u_ → 21a_1g_)^||^ and (1e_u_ → 14b_2g_)^||^ ones, while the opposite is true when L_3_^3^ is considered. In fact, once again in agreement with the experimental results of Kroll et al. [34], the electronic states determined by ΔS = 0, 1 coupled-single and single ⊥ polarized electronic transitions provide to the L_3_ higher *EE* region a contribution significantly lower than those || polarized.

DFT/ROCIS calculations fairly reproduce both the L_2_–L_3_ Δ*EE* (~10 eV) and the corresponding relative intensities; nevertheless, any detailed assignment of the L_2_ feature is herein avoided as it is not unambiguously determined by experimentation [32,80].

*FePc L_3_-edge.* FePc has been the object of a huge number of L_2,3_-edges XAS studies [33,34,38,120,124,125,126,127]. Among them, Bartolomé et al. [125] investigated the Fe magnetic moment switching in the catalytic ORR of FePc adsorbed on Ag(110) by combining the results of X-ray linear polarized absorption spectroscopy with those of X-ray magnetic circular dichroism at the Fe L_2,3_-edges. A detailed analysis of the || and ⊥ Fe L_2,3_-edges components of the FePc and FePc(*η*^2^-O_2_) XA spectra has been recently reported by Carlotto et al. [38] by adopting a computational set-up slightly different from that herein employed. The adopted set of c_1_, c_2_, and c_3_ semi-empirical parameters corresponded to that herein labelled old set; moreover, the saturation of the final-state manifold was obtained by considering forty nonrelativistic roots per multiplicity.

The comparison of the FePc ^||^^/^^⊥^*f*(*EE*) distributions (see Appendix A of the Appendix A) with the literature’s theoretical results [38] proves an evident disagreement. To disentangle the effects induced by the adoption of a particular set of semi-empirical parameters from those generated by the number of nonrelativistic roots per multiplicity, the ^||^^/^^⊥^*f*(*EE*) distributions have been again evaluated by using the c_1_, c_2_, and c_3_ old set (see Figure 4d). Besides minor differences, most likely due to the higher number of states and expansion vectors herein adopted in the iterative solution of the CI equations, the ^||^^/^^⊥^*f*(*EE*) distributions obtained by adopting c_1_ = 0.21, c_2_ = 0.49, and c_3_ = 0.29 substantially mirror the literature’s ones [38].

According to Carlotto et al. [38], weighty contributions to the ^||^^/^^⊥^*f*(*EE*) distributions arise from states having a spin multiplicity either equal to (ΔS = 0) or higher/lower (ΔS = ±1) than the GS one. More specifically, the ^||^^/^^⊥^L_3_^1^ features lying at the lowest excitation energy (see Figure 4d) are both associated to triplet states (ΔS = 0), and are generated by single electronic excitations with the whole Fe 2p set as *isos* and the low-lying Fe 3d-based 21a_1g_^↓^ VMO and 6e_g_^↓^ SOMO as *fsos*. Perfectly in agreement with the well-evidenced experimental dichroism [34,38], ^⊥^L_3_^1^ is much more intense than ^||^L_3_^1^, thus indicating that electronic states associated with (a_2u_ → a_1g_)^⊥^/(e_u_ → e_g_)^⊥^ excitations contribute much more than the (a_2u_ → e_g_)^||^/(e_u_ → a_1g_)^||^ ones to the lower *EE* region of the L_3_-edge. Excitations with ΔS = 0, −1 comparably participate in states determining ^⊥^L_3_^2^ (the theoretical set-up herein adopted makes negligible ΔS = 1 contributions). As such, ΔS = 0 electronic excitations involve the Fe-based 6e_g_^↓^ and 21a_1g_^↓^ VMO, while the ΔS = −1 ones have an MLCT character and imply Pc-based *π*^*^ VMOs. Moving to the analysis of the ^||^L_3_^2^ and ^||^L_3_^3^ features, the comparison with the XAS evidence indicates that their excitation energies are slightly overestimated with respect to the ^||^L_3_^1^ one; moreover, electronic states with ΔS = 0 (71%), 1 (19%) and −1 (7%) contribute to them. In even more detail, both single (44%) and coupled-single (56%) excitations contribute to the prevailing ΔS = 0 states. The former involves Fe-based (2p → 21a_1g_^↓^/6e_g_^↓^, 2p → 16b_1g_) and MLCT (2p → 7e_g_) transitions, while the latter are Fe-based (2p → SOMOs and SOMOs → 16b_1g_) excitations, determining the states contributing to the ^||^L_3_^2^ and ^||^L_3_^3^ lower excitation energy sides.

The DFT/ROCIS calculations [38] fairly reproduce both the L_2_–L_3_ Δ*EE* (~13 eV) and the corresponding relative intensities; nevertheless, any detailed assignment of the L_2_ feature is herein avoided as it is not unambiguously determined by experimentation [32,80].

## 4. Conclusions

The occupied and empty states of HS VPc, CrPc, IS MnPc and FePc have been thoroughly investigated by exploiting the original/homogeneous theoretical results and experimental evidence form the literature. The use of the Hirshfeld charges [84] coupled with the Nalewajski−Mrozek [66,67,68,69,70,71] approach ultimately indicates that, among the investigated molecules, FePc is characterized by the strongest and most covalent M–Pc *σ* interaction. Even though Slater’s transition state calculations ultimately confirm the Berkowitz hypothesis that, for FePc, “…the first ionization potential corresponds to electron ejection from a ring orbital, and not a metal orbital’’, the extension of the method to lighter MPcs reveals significant differences, the rationale of which lies with the relative energy position of the Pc^2−^-based and M^II^ 3d-based occupied/half-occupied MOs. Insights into the MPcs’ virtual electronic structure have been gained by revisiting XAS data from the literature in light of DFT/ROCIS calculations. The higher *EE* side of the MPc L_3_-edge XA spectra systematically includes states associated with MLCT transitions, most of them involving the metal 2p → Pc^2^^−^-based 7e_g_
*π*^*^ VMO excitations; moreover, the same *EE* region of all but one of (VPc) the L_3_-edge XA spectra is characterized by the presence of electronic states associated with M-based 2p → 16b_1g_ excitations. The agreement between theory and experiment is satisfactory, but it required a “tuning” of the modelling set-up in terms of XC functionals and/or c_1_, c_2_, and c_3_ semiempirical parameters. As a final consideration, we underline that the theoretical outcomes obtained for the HS MPc (VPc and CrPc) are a true challenge for the experimental community called upon to confirm or deny them.

## Figures and Tables

**Figure 1 nanomaterials-11-00054-f001:**
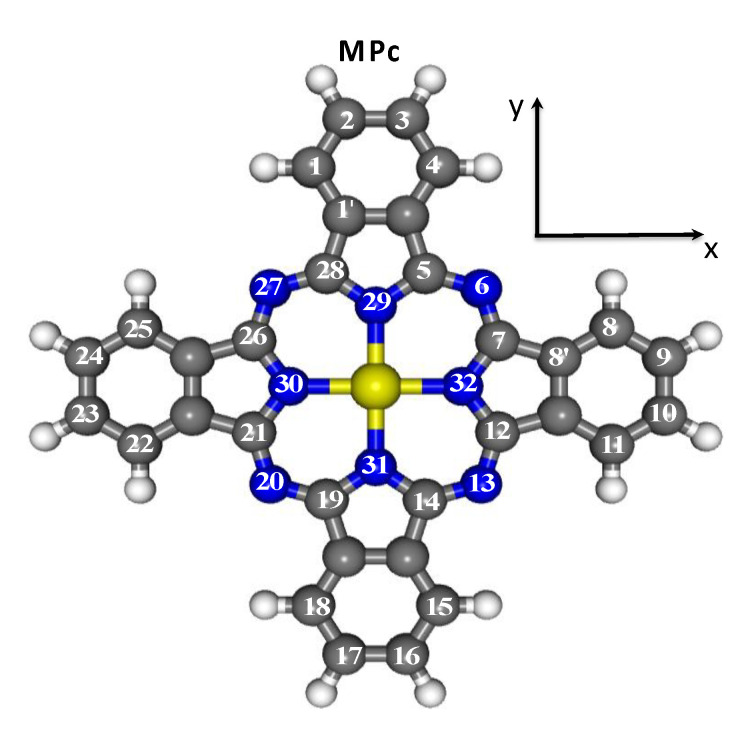
Schematic representation of a D_4h_ MPc molecule with the atom numbering recommended by the International Union of Pure and Applied Chemistry (IUPAC). White, grey, blue and yellow spheres correspond to H, C, N, and M atoms, respectively. In the adopted framework, the molecular *σ*_h_ plane corresponds to the xy plane. N(29), N(30), N(31) and N(32) (N(6), N(13), N(20) and N(27)) are collectively labelled N^Py^ (N^m^).

**Figure 2 nanomaterials-11-00054-f002:**
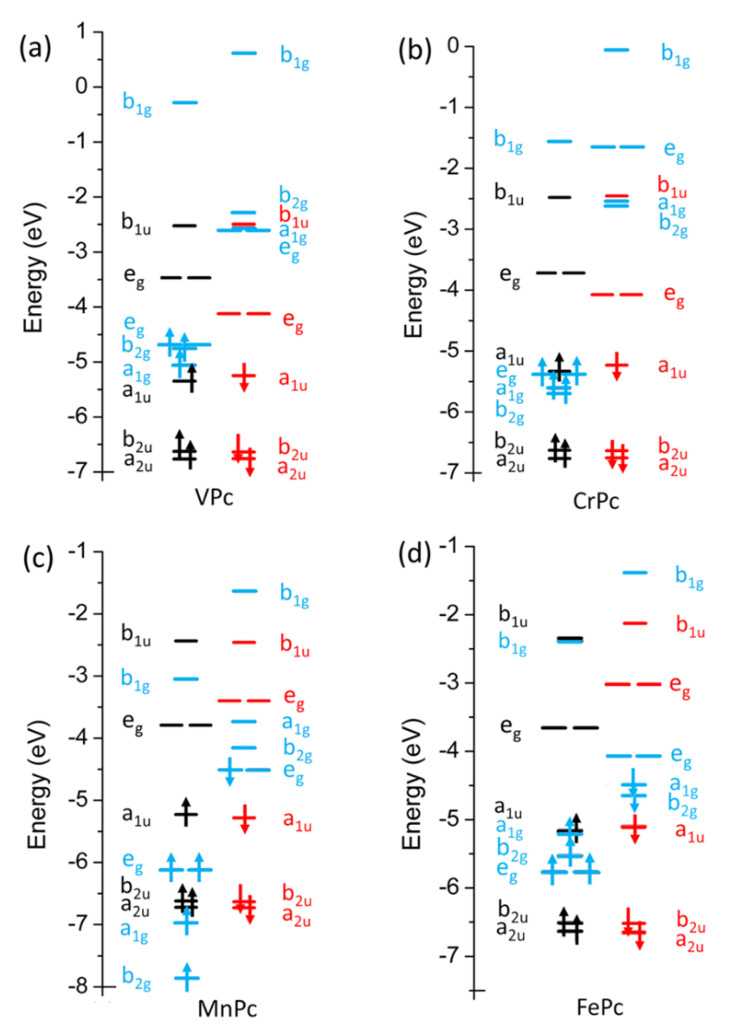
Relative energy positions of VPc (**a**), CrPc (**b**), MnPc (**c**) and FePc (**d**) frontier MOs. Black (↑)/red (↓) arrows refer to Pc-based selected orbitals, while the blue ones correspond to the M 3d-based MOs.

**Figure 3 nanomaterials-11-00054-f003:**
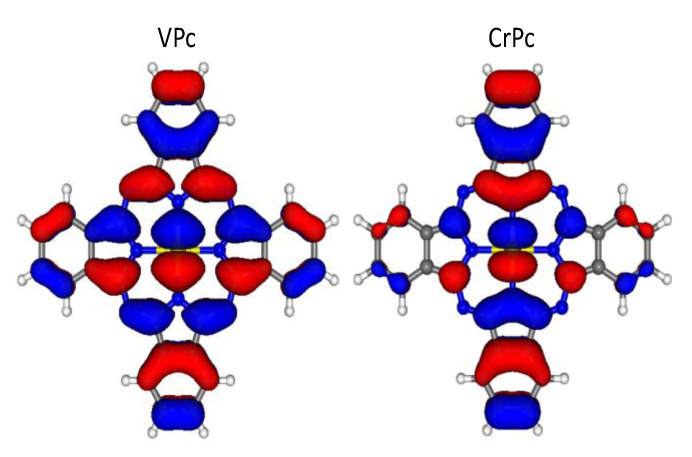
Three-dimensional plots of the VPc and CrPc 6e_g_^↑^ MO. Displayed isosurfaces correspond to ± 0.015 e^1/2^A^−3/2^.

**Figure 4 nanomaterials-11-00054-f004:**
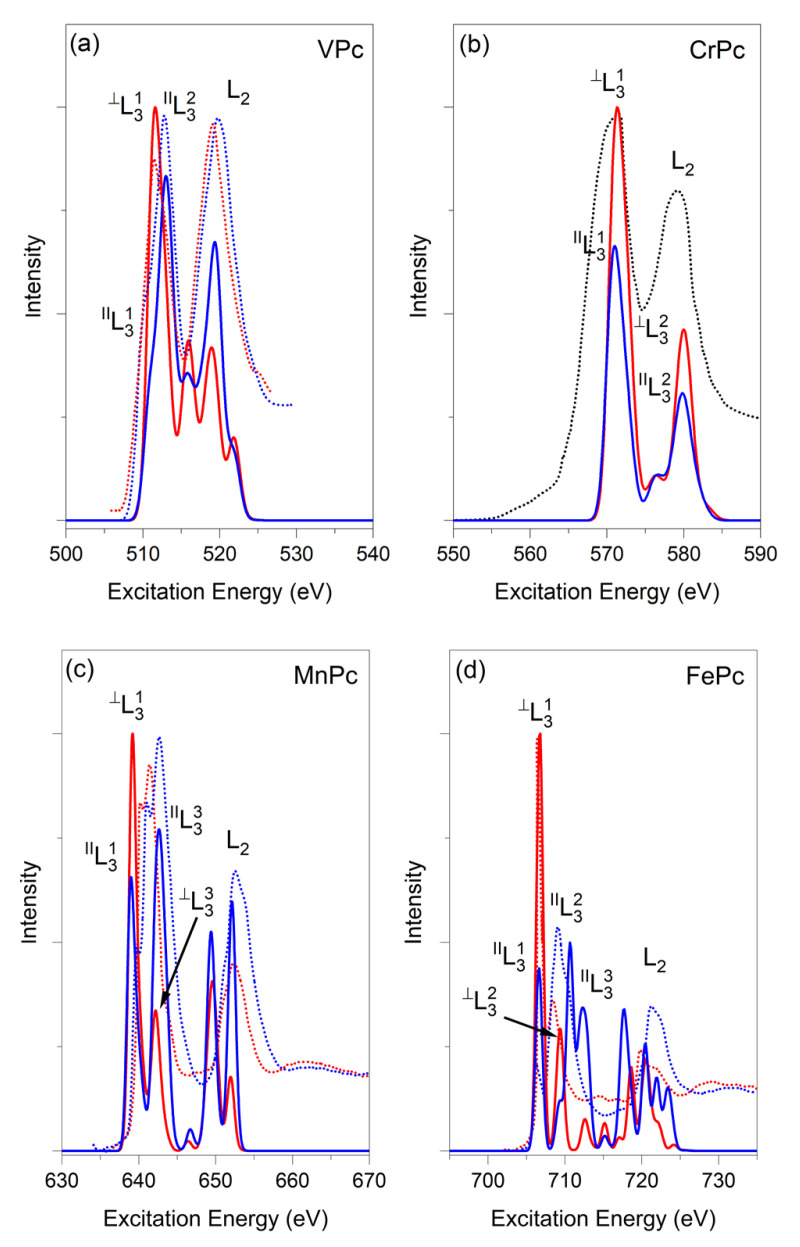
Experimental (dotted lines) and calculated (solid lines) ^||^^/^^⊥^*f*(*EE*) distributions for ^HS^VPc (**a**), ^HS^CrPc (**b**), ^IS^MnPc (**c**) and ^IS^FePc (**d**). Blue and red lines correspond to || and ⊥ components, respectively. Simulated spectra have been shifted by 10.9 (^HS^VPc), 9.8 (^HS^CrPc), 8.3 (^IS^MnPc) and 13.1 (^IS^FePc) eV and have a Gaussian broadening of 1.8 (^HS^VPc), 2.0 (^HS^CrPc), 1.0 (^IS^MnPc) and 1.0 (^IS^FePc) eV. Only the total experimental spectrum (black dotted line) is available for CrPc [33]; MnPc ^||^^/^^⊥^*f*(*EE*) distributions have been obtained by using the hybrid M06 meta-GGA XC [75] (see text).

**Table 1 nanomaterials-11-00054-t001:** BP86 Δ*BE* (kcal/mol) of optimized D_4h_ VPc, CrPc, MnPc and FePc with different spin states. GSs are taken as reference.

S	VPc	CrPc	MnPc	FePc
0		NC ^a^		31.9 (^1^A_1g_) ^b^
1/2	16.6 (^2^B_1g_)		14.9 (^2^B_2u_)	
1		25.6^b^ (^3^E_u_)		0 (^3^A_2g_)^90^
3/2	0 (^4^E_g_)		0 (^4^E_g_)	
2		0 (^5^B_1g_)		NC^a^
5/2			16.5 (^6^E_g_)	

^a^ NC stands for non-converged; ^b^ Non-Aufbau.

**Table 2 nanomaterials-11-00054-t002:** Theoretical and experimental [88] (in parentheses) structural parameters for D_4h_ VPc, CrPc, MnPc and FePc. Bond lengths/bond angles in Å/°, respectively.

	M–N^Py^	N^Py^–C	N*^m^*–C	M–N^Py^–C	N^Py^–C–N*^m^*
VPc ^*^	1.996	1.392	1.333	125.4	127.4
CrPc ^*^	1.982	1.387	1.330	125.6	127.5
MnPc ^a^	1.952 (1.938)	1.396 (1.392)	1.324 (1.315)	126.1 (126.2)	127.4 (127.6)
FePc ^b^	1.935 (1.927)	1.393 (1.378)	1.321 (1.322)	126.4 (126.3)	127.3 (127.8)

* Neither VPc nor CrPc crystallographic data are available in the literature; ^a^ from Ref. [92]; ^b^ from Ref. [93].

**Table 3 nanomaterials-11-00054-t003:** BP86 $\Delta E_{orb}^{\chi }$ (kcal/mol) of GS D_4h_ MPc herein considered (FePc is taken as a reference).

	VPc	CrPc	MnPc	FePc
a_1g_	100.9	106.7	103.8	0.0
b_1g_	−86.8	−85.9	−26.8	0.0
b_2g_	81.6	99.6	86.2	0.0
e_g_	−16.7	−55.0	−153.9	0.0
ΔE_orb_	79.0	64.8	9.3	0.0

## Data Availability

All data have been illustrated in the manuscript and in the supplementary materials.

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
