# Peer review of "A Theoretical Study of the Occupied and Unoccupied Electronic Structure of High- and Intermediate-Spin Transition Metal Phthalocyaninato (Pc) Complexes: VPc, CrPc, MnPc, and FePc"

_nanomaterials, 2020, doi:10.3390/nano11010054_

Round 1

Reviewer 1 Report

Carlotto and coworkers show here a DFT based study aimed at understanding the  excitation energies of a series of metal phthalocyaninato complexes, comparing to experiments, and serving as a way to understand the chemical bonding in such systems. The study is interesting, well carried out, well explained, and, other that very few typos, there is no criticism on that. I am glad to recommend publication in its present form.

Author Response

We thank the Referee for his/her gratifying evaluation of the manuscript.

Reviewer 2 Report

This manuscript reports a theoretical DFT study of the occupied and unoccupied electronic structure and high-, intermediate- and low-spin transitions in the metal phthalocyaninato complexes: VPc, CrPc, MnPc, and FePc. The complexes of this kind are widely used in various applications: in dyestuffs, semiconductors, chemical sensors, organic light emitting diodes, photovoltaic cells, thin-film transistors, materials for nonlinear optics, spintronics, laser recording etc. Any further clarification of the mechanisms behind their properties may give a substantial improvement of applications in these areas.

The authors calculated the properties of these complexes using the Amsterdam Density Functional (ADF) package and GGA / B3LYP / ROCIS methods, calculated the structural parameters, relative energy positions of VPc, CrPc, MnPc and FePc frontier MOs. The Ziegler’s extended transition state method and the DFT results allowed the authors to carry an advanced charge density and bond order analysis. It was found that the M-Pc bonding is dominated by interactions, with FePc having the strongest and most covalent M-Pc bond. The calculated structural parameters for FePc and MnPc are in a good agreement with the experimental ones. The experimental and calculated distributions for L2 and L3 spectra are found in the overall agreement in parallel and perpendicular components. 

This manuscript provides a comprehensive analysis of the calculated electronic spin states and L2 and L3 spectra with respect to the previous calculations and other experimental data and interpretations. It deserves publication in Nanomaterials because it provides new information based on the original research. The manuscript is very well-organized and written. It has supplementary materials, more than 100 relevant references and comments.  

There are several points which I recommend to clarify in the text or change: 

1) In Introduction, lines 1-69 are very general, this part can be supplemented by some specific references for VPc, CrPc, MnPc, and FePc, applications of these complexes and SO. 

2) On the other hand, in the discussion of results, e.g., lines 236-238, one can find that "FePc is broadly employed as a cheap and stable catalyst in several syntheses [9] including N-alkylation [85a], C-H amination [85b], C-C bond formation [85c], synthesis of esters [85d] and oximes [85e] as well as in reduction [85f], oxidation [85g-h], and radical reactions [85i]." This and similar parts are more appropriate in the introduction section. 

3) In Figure 4c and lines 373-401, the authors state that the hybrid M06 meta-GGA XC results are given and compared with the experimental data instead of the B3LYP/ROCIS results. The details of these calculations should be discussed in Section 2. Computational Details. 

After these minor changes, the manuscript can be published in Nanomaterials.

Author Response

We thank the Referee for his/her gratifying evaluation of the manuscript. As far as his/her comments are concerned, the description of peculiar CrPc, MnPc and FePc applications has been moved from the Results and Discussion section (Section 3) to the Section 1: Introduction. The same holds for the adoption of the M06 meta-GGA exchange-correlation potential to model the MnPc L2,3-edges XA spectra whose mentioning has been moved from Section 3 to the Section 2: Computational Details.

Reviewer 3 Report

The manuscript report DFT calculations of the geometry and electronic structure as well as X-ray absorption spectra with DFT/ROCIS of 4 important MPc (M=V, Cr, Mn, Fe) molecules. The previous calculations and various experimental studies are discussed in good detail. I find the paper carefully written and interesting and deserves publication.

Typo: "syn" -> "sym" in eq. (3).

Author Response

We thank the Referee for his/her gratifying evaluation of the manuscript.The typo in the eq. (3) has been fixed.